# Development and validation of a new prognostic index for mortality risk in multimorbid adults

**Viktoria Gastens**[1,2,3]\*, **Arnaud Chiolero**[1,3,4], **Daniela Anker**[3], **Claudio Schneider**[5], **Martin Feller**[1,5], **Douglas C. Bauer**[6], **Nicolas Rodondi**[1,5], **Cinzia Del Giovane**[1,3]

**1** Institute of Primary Health Care (BIHAM), University of Bern, Bern, Switzerland, **2** Graduate School for Health Sciences, University of Bern, Bern, Switzerland, **3** Department of Community Health, Population Health Laboratory (#PopHealthLab), University of Fribourg, Fribourg, Switzerland, **4** School of Population and Global Health, McGill University, Montreal, Canada, **5** Department of General Internal Medicine, Inselspital, Bern University Hospital, University of Bern, Bern, Switzerland, **6** Departments of Medicine and Epidemiology & Biostatistics, University of California, San Francisco, CA, United States of America

\* viktoria.gastens@biham.unibe.ch

## Abstract

### Context

Multimorbidity is highly prevalent among older adults and associated with a high mortality. Prediction of mortality in multimorbid people would be clinically useful but there is no mortality risk index designed for this population. Our objective was therefore to develop and internally validate a 1-year mortality prognostic index for older multimorbid adults.

### Methods

We analysed data of the OPERAM cohort study in Bern, Switzerland, including 822 adults aged 70 years or more with multimorbidity (3 or more chronic medical conditions) and polypharmacy (use of 5 drugs or more for >30 days). Time to all-cause mortality was assessed up to 1 year of follow-up. We performed a parametric Weibull regression model with backward stepwise selection to identify mortality risk predictors. The model was internally validated and optimism corrected using bootstrapping techniques. We derived a point-based risk score from the regression coefficients. Calibration and discrimination were assessed by the calibration slope and C statistic.

### Results

805 participants were included in the analysis. During 1-year of follow-up, 158 participants (20%) had died. Age, Charlson-Comorbidity-Index, number of drugs, body mass index, number of hospitalizations, Barthel-Index (functional impairment), and nursing home residency were predictors of 1-year mortality in a multivariable model. Using these variables, the 1-year probability of dying could be predicted with an optimism-corrected C statistic of 0.70. The optimism-corrected calibration slope was 0.93. Based on the derived point-based risk score to predict mortality risk, 7% of the patients classified at low-risk of mortality, 19% at moderate-risk, and 37% at high-risk died after one year of follow-up. A simpler mortality

**Data Availability Statement:** This study involves human research participant data containing sensitive patient information. In the EU Horizon 2020 grant agreement for the OPERAM study, it

had been specified that the data will be made available upon request if the use has been approved by an ethical committee. Therefore, restrictions to make the underlying data directly publicly available are both due to legal and ethical reasons, as health data are sensitive data. Data for this study will be made available for scientific purposes upon request for researchers whose proposed use of the data has been approved by the OPERAM publication committee. After approval and signing of a data transfer agreement ensuring adherence to privacy and data handling, data and documentation will be made available through a secure file exchange platform. Partially deidentified participant data, a data dictionary and annotated case report forms will be made available. For data access, external researchers can fill in the contact form on https://www.operam-cohort.biham.ch/ or operam-2020.eu.

**Funding:** This work was supported by Swiss National Science Foundation grants (320030_188549/01 to AC; 325130_204361 to CDG; www.snf.ch). This work is part of the project "OPERAM: OPtimising thERapy to prevent Avoidable hospital admissions in the Multimorbid elderly" supported by the European Union's Horizon 2020 research and innovation programme under the grant agreement No 6342388 (ec. europa.eu/programmes/horizon2020), and by the Swiss State Secretariat for Education, Research and Innovation (SERI) under contract number 15.0137 (www.sbfi.admin.ch/sbfi/en/home.html). The opinions expressed and arguments employed herein are those of the authors and do not necessarily reflect the official views of the EC and the Swiss government. The funders had no role in study design, data collection and analysis, decision to publish, or preparation of the manuscript.

**Competing interests:** The authors have declared that no competing interests exist.

score, without the Charlson-Comorbidity-Index and Barthel-Index, showed reduced discriminative power (optimism-corrected C statistic: 0.59) compared to the full score.

## Conclusion

We developed and internally validated a mortality risk index including for the first-time specific predictors for multimorbid adults. This new 1-year mortality prediction point-based score allowed to classify multimorbid older patients into three categories of increasing risk of mortality. Further validation of the score among various populations of multimorbid patients is needed before its implementation into practice.

## Introduction

Multimorbidity is highly prevalent especially in older adults [1] and, due to the population ageing, the number of people with multimorbidity is growing rapidly [2, 3]. Multimorbidity is associated with a high mortality rate [2] and previous research suggests that all-cause mortality risk for individuals with multimorbidity is nearly 2–3 times higher compared with individuals without multimorbidity [4–6].

Many guidelines recommend tailoring preventive care of multimorbid people according to life expectancy, and hence on the mortality risk [7]. Indeed, patients with high short-term mortality might not have the time to benefit from a preventive care intervention. Because multimorbid patients have a relatively short life expectancy, they are at higher risk of not having the time to benefit. It is therefore necessary to have a valid index for mortality prediction in multimorbid patients.

In a systematic review, Yourman et al. have identified 16 mortality prognostic indices for older adults [8]. While several of these prognostic indices were fairly accurate to predict mortality, the authors concluded that none could be recommended for a widespread use. One major limitation is that none of these prognostic indices has been tested prospectively in various samples. Key is that their transportability in other populations is unknown and clinicians cannot use these indices with confidence across different groups of patients [8]. Further, none of these indices has been developed specifically in multimorbid older adults. Current mortality indices do not consider predictors specific to multimorbid older adults such as the severity of comorbidities and functional impairment. Therefore, there is no tool recommended to accurately predict mortality in older multimorbid adults.

Our objective was therefore to develop and internally validate a 1-year mortality prognostic index for older multimorbid adults.

## Methods and analysis

### Source of data and study design

We used data from 822 participants of an ongoing cohort study in Bern, Switzerland. Participants were originally enrolled in the clinical trial OPtimising thERapy to prevent Avoidable hospital admissions in Multimorbid older people (OPERAM [9, 10]). Participants were assigned to receive either standard care or a medication review by a Systematic Tool to Reduce Inappropriate Prescribing (STRIP) with observation of the primary outcome of drug-related hospital admission (DRA) over 1 year.

For the current study, we used data collected at baseline (December 2016-October 2018) and up to 1 year after baseline (until October 2019). The local Ethics Committee in Bern, Switzerland, approved the study protocol with the project number 2018–00784. Study nurses collected baseline data by a personal interview with the participant and from medical files. The follow-up was conducted via phone calls. Phone interviews were held with participants or relatives, otherwise with a proxy or with the general practitioner, when the participants were not reachable or not able to answer.

We developed and validated the mortality prognostic index following the Prognosis Research Strategy (PROGRESS) framework [11], and reported it following the Transparent Reporting of a multivariable prediction model for Individual Prognosis Or Diagnosis (TRIPOD) statement [12, 13]. We further followed the recommendations of Moons et al. [14, 15] for risk prediction models. This study is part of a research project whose protocol has been published previously [10].

## Participants

Participants (N = 822) were enrolled at the time of a hospitalization in the Inselspital, University Hospital, Bern, Switzerland. Inclusion criteria were age of 70 years or older, multimorbidity (3 or more chronic medical conditions), and polypharmacy (use of 5 drugs or more for >30 days).

Written informed consent by patients themselves or, in the case of cognitive impairment by a legal representative, had already been obtained before enrolment. Patients planned for direct admission to palliative care (<24 hours after admission), or patients undergoing a structured drug review other than the trial intervention, or who had passed a structured drug review within the last 2 months were excluded. Patients for whom it was not possible to obtain an informed consent were excluded.

## Outcome

The outcome was time to all-cause mortality over one year of follow-up. Information on death and relative date was collected by study nurses through follow-up calls or primary care physician contact.

## Candidate predictors identification

Candidate 1-year mortality predictors were derived from previous research efforts in this field [8, 16], ease and reliability of measurement in clinical setting, and background knowledge on potential associations with mortality. We also considered factors included in the OPERAM dataset that may not be identified from the literature but are specific to multimorbid patients. All candidate predictors were based on baseline characteristics. We included demographic variables (age, sex), clinical characteristics (Charlson-Comorbidity-Index, number of drugs, body mass index, weight loss during the last year), smoking, functional status variables (Barthel-Index, falls, nursing home residence), and hospitalization. The variables about falls and hospitalization reflected the number of events during the last 12 months before index hospitalization. The Charlson-Comorbidity-Index was originally developed to predict mortality, with higher scores indicating higher mortality risk. It was calculated using ICD-10 codes by the adaptation of Quan et al. and implemented in the *comorbidity* library for the R environment [17, 18]. The Barthel-Index measures performance in activities of daily living (ADL) on an ordinal scale from 0 to 100 with higher scores indicating more independence. Continuous variables were categorised for ease of use at the point of care [19]. We based the categorisation

decisions on the analysis of the frequency distribution of the variables and on clinical rationale.

## Statistical analysis

We have calculated the required sample size for conducting our multivariable prediction model utilizing the criteria proposed by Riley et al. [20] and implemented in the *pmsampsize* library for the R environment [21]. The minimum sample size required with 12 candidate predictor parameters, an expected outcome event rate of 0.15 per year, and an anticipated Cox-Snell $R^2$ of 0.126 (C statistic of [16] 0.82, [22]) is 799 with 10 events per predictor parameter. Our sample size of 822 is therefore adequate for this project.

The relationship between candidate predictors and outcome was analysed using a parametric Weibull regression model. We first performed a univariable analysis between each potential candidate predictors and the outcome. We then used multiple imputation (number of multiple imputations, m = 10) for missing values under a missing at random assumption in order to reduce bias and avoid excluding participants from the analysis [14]. We performed stepwise backwards variable selection based on the Akaike Information Criterion (AIC) in each imputed dataset. For automatic selection procedures, backward elimination is recommended in TRIPOD [13, 23]. We started with the full model, sequentially dropping variables to maximally reduce AIC. The final model was formed of predictors for which AIC cannot be minimized further and which appeared in all models of the imputed datasets. The final model was fitted in each imputed dataset and results pooled according to Rubin's rules. We used the *mice* library for multiple imputation and pooling, and the *MASS* library with the *stepAIC* function for stepwise model selection via AIC in the R environment [24, 25].

We investigated the predictive accuracy of the final model by testing calibration and discrimination. The apparent performance and discrimination of the model was assessed with C statistic [14]. We evaluated potential overfitting and optimism by internal validation with bootstrapping techniques [14, 15]. We performed 500 bootstrap cycles. In each bootstrap sample, we derived a mortality prediction model and the relative risk score, as done in the original sample. We calculated optimism as difference in performance measure (C statistic) between the original sample and the respective bootstrap sample. This was repeated for all bootstrap samples to estimate the average optimism. We evaluated the calibration slope and intercept (calibration-in-the-large). We assessed graphical discrimination with a Kaplan-Meier plot of the risk groups.

**Point-based risk score.** From the final model, we derived a point-based risk score by assigning points to each risk factor. Each $\beta$ coefficient was divided by the lowest $\beta$ coefficient and rounded to the nearest integer. We calculated the total risk score for each participant by summing the points for each risk factor [16].

**Sensitivity analyses.** We performed a flexible parametric model to assess the robustness of the results obtained with the Weibull regression model. We performed a univariable analysis of the specific items in the Charlson-Comorbidity- Index and Barthel-Index. Further, to develop a score easier to use in practice, we performed the described methods above to develop and test a simplified model without the Barthel-Index and Charlson-Comorbidity-Index, because the assessment of both these indices can be difficult at the point-of-care.

## Results

Among 822 participants, 805 were included in the analytical sample. We excluded 17 participants because they left the study, and most data were missing. Baseline characteristics of the participants are reported in Table 1. The mean (min to max) age of participants was 79.7 (70

**Table 1. Baseline characteristics of all participants and of those who died during follow-up.**

| Variables | | Total (%) | Deaths (%) |
|---|---|---|---|
| | | n = 805[a] | n = 158 |
| Age | 70–79 | 421 (52) | 69 (44) |
| | 80–99 | 384 (48) | 89 (56) |
| Sex | Female | 338 (42) | 68 (43) |
| | Male | 467 (58) | 90 (57) |
| CC-Index | 0–2 | 319 (40) | 35 (22) |
| | ≥3 | 486 (60) | 123 (78) |
| Drugs[b] | <10 | 359 (45) | 55 (35) |
| | ≥10 | 446 (55) | 103 (65) |
| BMI | <30 | 611 (76) | 126 (80) |
| | ≥30 | 164 (20) | 22 (14) |
| Weight loss[§] | Yes | 255 (32) | 59 (37) |
| | No | 545 (68) | 98 (62) |
| Smoking | Yes | 69 (9) | 14 (9) |
| | No | 733 (91) | 144 (91) |
| Hospitalizations[c] | 0 | 386 (48) | 63 (40) |
| | ≥1 | 416 (52) | 95 (60) |
| Barthel-Index | <21 | 34 (4) | 18 (11) |
| | 21–60 | 144 (18) | 42 (27) |
| | 61–90 | 237 (29) | 53 (34) |
| | >90 | 377 (47) | 42 (27) |
| Falls[§] | 0 | 445 (55) | 78 (49) |
| | 1 | 174 (22) | 31 (20) |
| | >1 | 182 (23) | 49 (31) |
| Nursing home residence | Yes | 70 (9) | 15 (9) |
| | No | 735 (91) | 143 (91) |

[a]17 participants excluded (from 822 to 805) due to premature study end.

[b]before index hospitalization.

[c]during last 12 months.

to 99) years and 42% were women. During a mean (min to max) follow-up of 12.2 months (11.0 to 17.1), 158 participants (20%) had died. The proportion of missing data ranged from 0% to 10% in the predictor variables (S1 Table). The univariable analysis showed that age, Charlson-Comorbidity-Index, number of drugs, BMI, number of hospitalizations, and Barthel-Index were associated with 1-year mortality (S1 Table).

The final risk prediction model included the predictors age, BMI, Charlson-Comorbidity-Index, hospitalizations, drugs, Barthel-Index, and nursing home residence (Table 2). We generated 10 imputed datasets by multiple imputation for missing data. The final predictor variables were retained in all imputed datasets after stepwise selection. Table 3 showed apparent and internal validation performance statistics of our risk prediction model. After adjustment for optimism with bootstrapping, our final model was able to discriminate participants with and without death within one year with a C statistic of 0.70. The optimism-corrected calibration slope was 0.93.

## Sensitivity analyses

We found similar results by applying the flexible parametric model compared to the main analysis (S3 Table). Univariable analysis of the Charlson-Comorbidity-Index and the Barthel-

**Table 2. 1-year mortality predictors retained in the final model and associated risk score.**

| Variable | | HR (95% CI) | β coefficient[a] | p-value | Risk score |
|---|---|---|---|---|---|
| Age | 70–79 | Ref | Ref | | |
| | 80–99 | 1.30 (0.94–1.79) | 0.26 | 0.01 | 1 |
| CC-Index[b] | 0–2 | Ref | Ref | | |
| | ≥3 | 2.26 (1.55–3.31) | 0.82 | < .001 | 4 |
| Drugs | <10 | Ref | Ref | | |
| | ≥10 | 1.22 (0.87–1.72) | 0.20 | 0.25 | 1 |
| BMI[c] | ≥30 | Ref | Ref | | |
| | <30 | 1.67 (1.08–2.60) | 0.51 | 0.03 | 2 |
| Hospitalizations | 0 | Ref | Ref | | |
| | ≥1 | 1.28 (0.92–1.77) | 0.24 | 0.15 | 1 |
| Barthel-Index | >90 | Ref | Ref | | |
| | 61–90 | 12.00 (1.33–3.01) | 0.69 | < .01 | 3 |
| | 21–60 | 3.02 (1.96–4.65) | 1.10 | < .001 | 5 |
| | <21 | 7.79 (4.46–13.61) | 1.88 | < .001 | 9 |
| Nursing home residence | | 1.96 (1.13–3.42) | 0.67 | 0.02 | 3 |

[a] β coefficient = logHR.

[b] Charlson Comorbidity Index.

[c] Body-Mass-Index.

Index showed the potential of including specific items instead of the entire indices in further projects (S4 Table). A simplified risk score without the Barthel-Index and Charlson-Comorbidity-Index was developed. The risk score points assigned to each of the final predictors in this simpler risk score are listed in Table 4. The simplified model had a weaker discrimination performance with a C statistic of 0.59 (Table 5).

## Point-based risk score

The risk score points assigned to each of the final predictors are listed in Table 2. A risk score was calculated for each participant by adding the points for each predictor present. For example, a 81-year-old (1 points) woman with a CC-Index of 3 (4 points), taking 5 drugs (0 points), with a BMI of 28 (2 points), 2 hospitalizations in the last 12 months (1 point), a Barthel-Index of 80 (3 points), and living in a nursing home residence (3 points) would have a total risk score of 14 points.

The mean 1-year mortality risk score in our sample was 7.7 (standard deviation 3.9); it ranged from 2 to 21. Based on the derived point-based risk score to predict mortality risk, 7% (95% CI: 6.3–7.7) of patients in the low-risk category (0 to 5 points) died, 19% (11.4–24.6) with moderate-risk (6 to 10 points), and 37% (25.2–44.8) with high-risk of 1-year mortality (>10 points) (S5A Table). Fig 1A showed the Kaplan-Meier plot of the three risk groups and good graphical discrimination.

**Table 3. Apparent and internal validation performance statistics of the final prediction model (with 95% CI) including C statistic, Calibration slope and Calibration-in-the-large.**

| Performance measure | Apparent | Average optimism | Optimism corrected |
|---|---|---|---|
| C statistic | 0.71 | 0.02 | 0.70 (0.69–0.70) |
| C slope | 1 | 0.07 | 0.93 (0.92–0.94) |
| CITL | 0 | -0.61 | 0.61 (0.56–0.66) |

**Table 4. 1-year mortality predictors retained in the simplified model and associated risk score.**

| Variable | | HR (95% CI) | β coefficient[a] | p-value | Risks score |
|---|---|---|---|---|---|
| Age | 70–79 | Ref | Ref | | |
| | 80–99 | 1.41 (1.02–1.94) | 0.34 | 0.04 | 3 |
| Drugs | <10 | Ref | Ref | | |
| | ≥10 | 1.53 (1.09–2.14) | 0.43 | 0.01 | 3 |
| BMI[b] | ≥30 | Ref | Ref | | |
| | <30 | 1.61 (1.03–2.50) | 0.47 | 0.04 | 4 |
| Hospitalizations | 0 | Ref | Ref | | |
| | ≥1 | 1.39 (1.00–1.92) | 0.33 | 0.05 | 3 |
| Nursing home residence | | 1.13 (0.66–1.94) | 0.13 | 0.65 | 1 |

[a]β coefficient = logHR.

[b]Body-Mass-Index.

Based on the simpler risk score to predict mortality risk, 13% (95% CI: 10.9–15.1) of patients in the low-risk category (0 to 7 points) died, and 25% (20.1–29.9) with high-risk of 1-year mortality (>7 points) (S5B Table). Fig 1B showed the Kaplan-Meier plot of two risk groups of this simplified risk score.

## Discussion

We have developed and internally validated a new risk prediction model to estimate the 1-year mortality risk in older multimorbid adults. The final model included the seven predictors age, BMI, Charlson-Comorbidity-Index, number of hospitalizations, number of drugs, Barthel-Index, and nursing home residence. Using this score, we could classify patients into categories of increasing risk of 1-year mortality with a substantial risk difference. A simpler score was able to categorize 1-year mortality risk with a weaker discriminative power.

We used high-quality data from a large prospective cohort study of multimorbid older adults. In contrast to existing mortality risk indices, our index focuses specifically on multi-morbid older patients and accounted for the severity of comorbidity (Charlson-Comorbidity-Index) and functional impairment (Barthel-Index) [16]. The interpretation of performance measures such as the C statistic depends on the clinical area. Other prognostic indices for 1-year mortality in older adults show similar discrimination performance, e.g. C statistic of 0.68 or 0.79 [26, 27]. We have applied a particularly robust methodological framework by following the research guidelines in this field, namely the Prognosis Research Strategy (PROGRESS) framework, and the Transparent Reporting of a multivariable prediction model for Individual Prognosis Or Diagnosis (TRIPOD) statement. Notably, we ensured adequate sample size, used multiple imputation for missing data, and applied bootstrapping techniques for internal validation [13, 23].

**Table 5. Apparent and internal validation performance statistics of the simplified prediction model (with 95% CI) including C statistic, Calibration slope and Calibration-in-the-large.**

| Performance measure | Apparent | Average optimism | Optimism corrected |
|---|---|---|---|
| C statistic | 0.60 | 0.02 | 0.59 (0.58–0.59) |
| C slope | 1 | 0.13 | 0.87 (0.85–0.88) |
| CITL | 0 | -0.99 | 0.99 (0.75–1.24) |

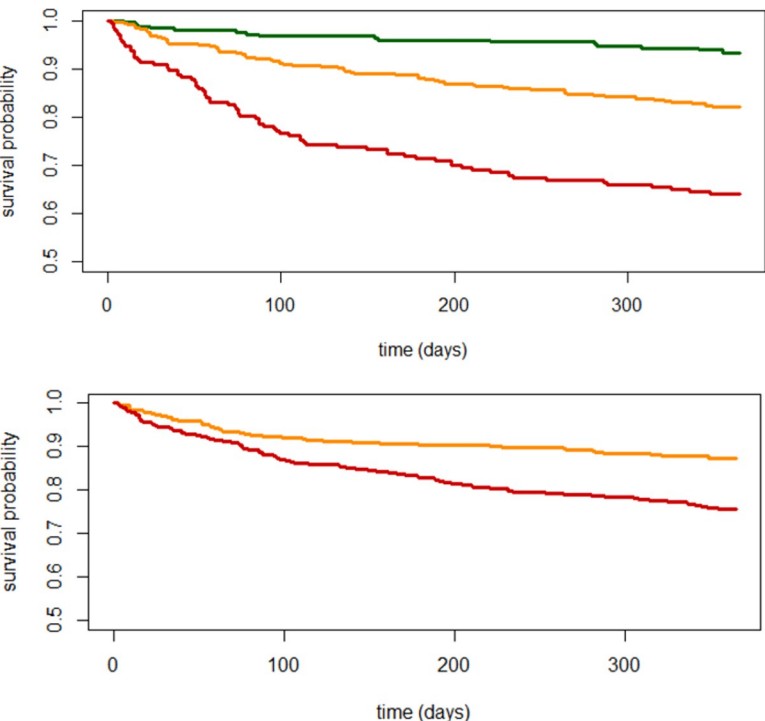

**Fig 1. A.** Kaplan-Meier curves in three risk groups to visually assess separation (low risk: 0–5 points (green), moderate risk: 6–10 points (orange), high risk: 11–21 (red)). **B.** Kaplan-Meier curves of the simplified score in two risk groups to visually assess separation (low risk: 0–7 points (orange), high risk: 8–14 (red)).

One major limitation is the lack of external validation but we will explore opportunities to test the score in a different dataset of older multimorbid adults. However, we did our best to assess internal validation by using the bootstrap method for internal validation that makes use of the entire sample. This optimism-corrected validation analysis indicates that both calibration slope and intercept deviate from the null value of one (0.93; 95% CI: 0.92, 0.94) and zero (0.61; 95% CI: 0.56, 0.66) (Table 3), respectively. The calibration slope evaluates the spread of the estimated risks and has a target value of 1. A slope < 1 suggests that estimated risks are too extreme, a slope > 1 suggests that risk estimates are too moderate [28]. The calibration intercept, which is an assessment of calibration-in-the-large, has a target value of 0; negative values suggest overestimation, whereas positive values suggest underestimation [28]. Therefore, our results indicate suboptimal prediction accuracy as the risk estimates were considered too extreme (for the calibration slope) and the model underestimating the predicted risk (for the calibration intercept). As the study participants were included at the time of hospitalization, they may not be representative of all patients with multimorbidity, and this could reduce the transportability of our findings to other populations. Another limitation is that indices such as the Barthel-Index and the Charlson-Comorbidity-Index used as predictor may reduce the ease of use of the risk score at the point-of-care. We have therefore developed a simpler score without such variables. This simpler score showed reduced discriminative power compared to the full score. This could highlight the importance of taking the severity of comorbidities and functional impairment into account to improve risk prediction in older multimorbid people. Additional studies will be needed to test the usefulness of both scores in practice. Finally, prognostic information longer than 1-year mortality risk is needed. We will expand this model to 3-year mortality risk once the 3-year follow-up data collection is completed [10]. For this next

research project, we might consider nomogram analysis as a graphically intuitive representation [29].

Our results will be useful for both clinical and research activities by helping health care providers to tailor preventive care according to the estimated mortality risk. Eventually, our study can help preventing under- and overuse of preventive care in the growing older population.

## Supporting information

**S1 Table. Missing data in the predictors in the 805 participants.**
(DOCX)

**S2 Table. Univariable analysis of candidate predictors with a Weibull model from information obtained at baseline.**
(DOCX)

**S3 Table. Multivariable analysis of candidate predictors with a flexible parametric survival model.**
(DOCX)

**S4 Table. Univariable analysis of the specific items in the Charlson Comorbidity Index and Barthel-Index.**
(DOCX)

**S5 Table A Stratification of 1-year mortality risk by points of the final score B Stratification of 1-year mortality risk by points of the simplified score.**
(DOCX)

**S1 Fig.** A. Calibration plot of the final prediction model for 1-year mortality. B. Calibration plot of the simplified model for 1-year mortality.
(TIF)

## Author Contributions

**Conceptualization:** Viktoria Gastens, Arnaud Chiolero, Daniela Anker, Nicolas Rodondi.

**Data curation:** Viktoria Gastens, Martin Feller, Nicolas Rodondi.

**Formal analysis:** Viktoria Gastens.

**Funding acquisition:** Arnaud Chiolero.

**Investigation:** Viktoria Gastens, Arnaud Chiolero, Claudio Schneider, Cinzia Del Giovane.

**Methodology:** Viktoria Gastens, Arnaud Chiolero, Nicolas Rodondi, Cinzia Del Giovane.

**Project administration:** Viktoria Gastens.

**Resources:** Viktoria Gastens.

**Software:** Viktoria Gastens.

**Supervision:** Arnaud Chiolero, Nicolas Rodondi, Cinzia Del Giovane.

**Validation:** Viktoria Gastens.

**Visualization:** Viktoria Gastens.

**Writing – original draft:** Viktoria Gastens, Arnaud Chiolero, Cinzia Del Giovane.

**Writing – review & editing:** Viktoria Gastens, Arnaud Chiolero, Daniela Anker, Claudio Schneider, Martin Feller, Douglas C. Bauer, Nicolas Rodondi, Cinzia Del Giovane.

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
