## [Decision Letter · Decision Letter 0]

15 Mar 2022

PONE-D-22-03428Development and validation of a new prognostic index for mortality risk in multimorbid adultsPLOS ONE

Dear Dr. Gastens,

Thank you for submitting your manuscript to PLOS ONE. After careful consideration, we feel that it has merit but does not fully meet PLOS ONE’s publication criteria as it currently stands. Therefore, we invite you to submit a revised version of the manuscript that addresses the points raised during the review process.

Both reviewers request fuller contextualization of your work.  Please see detailed reviewer comments.

We look forward to receiving your revised manuscript.

Kind regards,

Robert Daniel Blank, MD, PhD

Academic Editor

PLOS ONE

Journal Requirements:

"This work was supported by Swiss National Science Foundation grants (320030_188549/01 to AC; 325130_204361 to CDG). This work is part of the project “OPERAM: OPtimising thERapy to prevent Avoidable hospital admissions in the Multimorbid elderly” supported by the European Union's Horizon 2020 research and innovation programme under the grant agreement No 6342388, and by the Swiss State Secretariat for Education, Research and Innovation (SERI) under contract number 15.0137. The opinions expressed and arguments employed herein are those of the authors and do not necessarily reflect the official views of the EC and the Swiss government."

We note that you have provided funding information. However, funding information should not appear in the Funding section or other areas of your manuscript. We will only publish funding information present in the Funding Statement section of the online submission form. 

"This work was supported by Swiss National Science Foundation grants (320030_188549/01 to AC; 325130_204361 to CDG; www.snf.ch). This work is part of the project “OPERAM: OPtimising thERapy to prevent Avoidable hospital admissions in the Multimorbid elderly” supported by the European Union's Horizon 2020 research and innovation programme under the grant agreement No 6342388 (ec.europa.eu/programmes/horizon2020), and by the Swiss State Secretariat for Education, Research and Innovation (SERI) under contract number 15.0137 (www.sbfi.admin.ch/sbfi/en/home.html). The opinions expressed and arguments employed herein are those of the authors and do not necessarily reflect the official views of the EC and the Swiss government. The funders had no role in study design, data collection and analysis, decision to publish, or preparation of the manuscript."

Reviewers' comments:

Reviewer's Responses to Questions

**Comments to the Author**

1. Is the manuscript technically sound, and do the data support the conclusions?

Reviewer #1: Yes

Reviewer #2: Yes

2. Has the statistical analysis been performed appropriately and rigorously? 

Reviewer #1: Yes

Reviewer #2: Yes

3. Have the authors made all data underlying the findings in their manuscript fully available?

Reviewer #1: Yes

Reviewer #2: No

4. Is the manuscript presented in an intelligible fashion and written in standard English?

Reviewer #1: Yes

Reviewer #2: Yes

5. Review Comments to the Author

Reviewer #1: The authors sought to develop a risk assessment tool to predict 1-year mortality risk for elderly people with multimultimorbidity (i.e., 3+ chronic health diseases) and polypharmacy (i.e., use of 5+ drugs for 30+ days). This is a secondary analysis of 805 relevant participants from the OPERAM study, a European multicenter, cluster randomized controlled trial. Multiple imputation was performed for missing data, ranging from 0% to 10% under the assumption of missing at random. The predictor candidates were predefined and a backward stepwise selection was conducted to identify the optimal predictive model. Internal validation was carried out using bootstrapping techniques. The optimal model had the optimism-corrected C index of 0.70 and calibration slop of 0.93. A simpler model, excluding Charlson Comorbidity Index and Barthel Index with C index of 0.59 (95% CI: 0.58, 0.59) was considered of “satisfactory discriminative power”.

The manuscript is overall informative and well written. The study was well designed and executed, and the analysis was conducted appropriately. There are however several issues the authors might wish to consider in the subsequent submission.

1. Please clarify why a backward stepwise selection was used to identify the optimal selection model whereas the other methods, including Bayesian model averaging (BMA) method that have been demonstrated to be statistically more robust are widely available. Wang et al (https://pubmed.ncbi.nlm.nih.gov/15505893/) and Genell et al (https://pubmed.ncbi.nlm.nih.gov/21134252/) have found BMA was superior to stepwise method in model selection. Selection of a robust statistical method can be an important discussion issue to convince the findings were rigorous and accurate.

2. It is acceptable to report a calibration slope to assess the predictive performance of a prediction model. As a calibration slop evaluates the spread of the estimated risk, the manuscript should also report the calibration intercept, known as the “calibration-in-the-large” index which quantifies the overall calibration performance or the difference between the average predicted risk and the overall event rate.

3. It is also acceptable to derive a point-based risk score by assigning points to each predictor in the optimal model. However, the authors might wish to consider a more robust and friendly alternative approach, such as a nomogram analysis. Unlike a point-based risk score, a nomogram provides more graphically intuitive and more friendly risk estimation for subjects with specific risk profiles. A nomogram also provides a practical solution for the implementation progress and makes the manuscript more appealing.

4. A simpler model without Charson-Comorbidity index and Barthel index is considered satisfactorily discriminative though its C index was only 0.59. Please provide explanation and cite valid references for this statement as it is practically a bit hard, especially for people working in daily clinical practice to consider a discrimination index of 0.59 satisfactory. Further discussion is definitely needed.

5. The authors are expected to expand the Discussion section. For instance, they might wish to discuss (i) the robustness of the methods used in this study in comparison with previous studies or other methods, (ii) the results, including but not limited to a “satisfactory discriminative power” of 0.59, and (iii) potential implication of the model.

6. Minor issues:

The authors might wish to make their study rationale stronger and more appealing. For instance, they might add a sentence or two to extrapolate the reasons why several of 16 prognostic indices were “fairly accurate to predict mortality, the authors concluded that none could be recommended for a widespread use”.

Reviewer #2: Important topic, clear rationale of the study. I would like to congratulate the authors with a clearly written, and concise paper. I do have a few comments:

-the statistical analyses are sophisticated, and can be considered state-of-the-art in the field of health outcome predictions

-I do miss an independent population to assess external validity of the model

-The paper could be further strengthend by comparing performance of the model with other (simple) models that predict mortality. Is this model really more appropriate than those currently available?

-as the authors do acknowledge in the discussion, assessing performance in an extended time horizon beyond 1-year would be of interest

Thank you for having me as reviewer

6. PLOS authors have the option to publish the peer review history of their article (what does this mean?). If published, this will include your full peer review and any attached files.

Reviewer #1: **Yes: **Thach Tran

Reviewer #2: **Yes: **Silvan Licher

---

## [Author Response · Author response to Decision Letter 0]

28 Apr 2022

April, 2022

RE: PONE-D-22-03428 – “Development and validation of a new prognostic index for mortality risk in multimorbid adults”

To the Editor,

We thank you for your interest and reviewing our manuscript. We are grateful for the editor’s and reviewers’ comments which helped us to clarify and improve our manuscript. 

We have addressed all comments raised by the editor and the reviewers in the edited version of the manuscript and in our point-by-point responses to the reviewers' comments. All references quoted in our responses are listed at the end of this letter. We have uploaded a “Revised Manuscript with Track Changes” showing revisions using tracked changes. 

We hope that the revised version of our manuscript will be suitable for publication in PLOS ONE.

We are looking forward to hearing from you,

Kind regards

Viktoria Gastens, MSc

Institute of Primary Health Care (BIHAM), University of Bern

Population Health Laboratory (#PopHealthLab), University of Fribourg

Email: viktoria.gastens@biham.unibe.ch

 

Journal Requirements:

Authors’ response: We thank the editor for their comments below allowing us to improve our manuscript. 

Authors’ response: We have adapted the style of the manuscript according to the PLOS ONE formatting requirements.

"This work was supported by Swiss National Science Foundation grants (320030_188549/01 to AC; 325130_204361 to CDG). This work is part of the project “OPERAM: OPtimising thERapy to prevent Avoidable hospital admissions in the Multimorbid elderly” supported by the European Union's Horizon 2020 research and innovation programme under the grant agreement No 6342388, and by the Swiss State Secretariat for Education, Research and Innovation (SERI) under contract number 15.0137. The opinions expressed and arguments employed herein are those of the authors and do not necessarily reflect the official views of the EC and the Swiss government."

We note that you have provided funding information. However, funding information should not appear in the Funding section or other areas of your manuscript. We will only publish funding information present in the Funding Statement section of the online submission form. 

"This work was supported by Swiss National Science Foundation grants (320030_188549/01 to AC; 325130_204361 to CDG; www.snf.ch). This work is part of the project “OPERAM: OPtimising thERapy to prevent Avoidable hospital admissions in the Multimorbid elderly” supported by the European Union's Horizon 2020 research and innovation programme under the grant agreement No 6342388 (ec.europa.eu/programmes/horizon2020), and by the Swiss State Secretariat for Education, Research and Innovation (SERI) under contract number 15.0137 (www.sbfi.admin.ch/sbfi/en/home.html). The opinions expressed and arguments employed herein are those of the authors and do not necessarily reflect the official views of the EC and the Swiss government. The funders had no role in study design, data collection and analysis, decision to publish, or preparation of the manuscript."

Authors’ response: We have deleted the funding section in the manuscript. The funding information in the Funding Statement section of the online submission form is correct.

Authors’ response: We have updated the Data Availability statement accordingly: “This study involves human research participant data containing sensitive patient information. In the EU Horizon 2020 grant agreement for the OPERAM study, it had been specified that the data will be made available upon request if the use has been approved by an ethical committee. Therefore, restrictions to make the underlying data directly publicly available are both due to legal and ethical reasons, as health data are sensitive data. 

Data for this study will be made available for scientific purposes upon request for researchers whose proposed use of the data has been approved by the OPERAM publication committee. After approval and signing of a data transfer agreement ensuring adherence to privacy and data handling, data and documentation will be made available through a secure file exchange platform. Partially deidentified participant data, a data dictionary and annotated case report forms will be made available. For data access, external researchers can fill in the contact form on https://www.operam-cohort.biham.ch/ or operam-2020.eu.”

Authors’ response: We have moved the ethics statement to the Methods section of our manuscript.

Authors’ response: We have adapted the Supporting Information files and citations accordingly.

Authors’ response: We have reviewed our reference list. We have added references to: “Royston and Sauerbrei; Moons 2015”. This has led to a change in the order of references.

 

Reviewer 1

The authors sought to develop a risk assessment tool to predict 1-year mortality risk for elderly people with multimultimorbidity (i.e., 3+ chronic health diseases) and polypharmacy (i.e., use of 5+ drugs for 30+ days). This is a secondary analysis of 805 relevant participants from the OPERAM study, a European multicenter, cluster randomized controlled trial. Multiple imputation was performed for missing data, ranging from 0% to 10% under the assumption of missing at random. The predictor candidates were predefined and a backward stepwise selection was conducted to identify the optimal predictive model. Internal validation was carried out using bootstrapping techniques. The optimal model had the optimism-corrected C index of 0.70 and calibration slop of 0.93. A simpler model, excluding Charlson Comorbidity Index and Barthel Index with C index of 0.59 (95% CI: 0.58, 0.59) was considered of “satisfactory discriminative power”.

The manuscript is overall informative and well written. The study was well designed and executed, and the analysis was conducted appropriately. There are however several issues the authors might wish to consider in the subsequent submission.

Authors’ response: We thank the reviewer for their comments below allowing us to improve our manuscript. 

7. Please clarify why a backward stepwise selection was used to identify the optimal selection model whereas the other methods, including Bayesian model averaging (BMA) method that have been demonstrated to be statistically more robust are widely available. Wang et al (https://pubmed.ncbi.nlm.nih.gov/15505893/) and Genell et al (https://pubmed.ncbi.nlm.nih.gov/21134252/) have found BMA was superior to stepwise method in model selection. Selection of a robust statistical method can be an important discussion issue to convince the findings were rigorous and accurate.

Authors’ response: We agree with the reviewer that stepwise variable selection can have some disadvantages such as instability of selection or biased estimation. However, according to the Transparent Reporting of a multivariable prediction model for Individual Prognosis or Diagnosis (TRIPOD) statement, backwards elimination is still recommended as an automatic selection procedure. [Royston and Sauerbrei 2008; TRIPOD 2015] We have added to our Methods section: “If using automatic selection procedures, backward elimination is recommended in TRIPOD.”

8. It is acceptable to report a calibration slope to assess the predictive performance of a prediction model. As a calibration slop evaluates the spread of the estimated risk, the manuscript should also report the calibration intercept, known as the “calibration-in-the-large” index which quantifies the overall calibration performance or the difference between the average predicted risk and the overall event rate.

Authors’ response: We agree with the reviewer and have added the calibration intercept to the Results section in Table 3A and 3B.

9. It is also acceptable to derive a point-based risk score by assigning points to each predictor in the optimal model. However, the authors might wish to consider a more robust and friendly alternative approach, such as a nomogram analysis. Unlike a point-based risk score, a nomogram provides more graphically intuitive and more friendly risk estimation for subjects with specific risk profiles. A nomogram also provides a practical solution for the implementation progress and makes the manuscript more appealing.

Authors’ response: We agree with the reviewer that nomogram analysis is a graphically intuitive representation. However, nomograms represent models with continuous variables. Due to the binary and categorical characteristics of our predictor variables, a nomogram analysis is not applicable. For the next part of this research project (extending to 3-year mortality risk prediction), we plan to keep variables continuous wherever possible and could therefore apply a nomogram analysis. [Gastens 2021]

10. A simpler model without Charson-Comorbidity index and Barthel index is considered satisfactorily discriminative though its C index was only 0.59. Please provide explanation and cite valid references for this statement as it is practically a bit hard, especially for people working in daily clinical practice to consider a discrimination index of 0.59 satisfactory. Further discussion is definitely needed.

Authors’ response: We agree with the reviewer and revised the text as follows: “A simpler score was able to categorize 1-year mortality risk with a weaker discriminative power.” and “This simpler score showed reduced discriminative power compared to the full score.”

11. The authors are expected to expand the Discussion section. For instance, they might wish to discuss (i) the robustness of the methods used in this study in comparison with previous studies or other methods, (ii) the results, including but not limited to a “satisfactory discriminative power” of 0.59, and (iii) potential implication of the model.

Authors’ response: We have expanded our Discussion section about the robustness of the methods: “We have applied a particularly robust methodological framework by following the research guidelines in this field, namely the Prognosis Research Strategy (PROGRESS) framework, and the Transparent Reporting of a multivariable prediction model for Individual Prognosis Or Diagnosis (TRIPOD) statement. Notably, we ensured adequate sample size, used multiple imputation for missing data, and applied bootstrapping techniques for internal validation.” We further discuss the results: “In contrast to existing mortality risk indices, our index focuses specifically on multimorbid older patients and accounted for the severity of comorbidity (Charlson-Comorbidity-Index) and functional impairment (Barthel-Index).” We discuss potential implications: “Our results will be useful for both clinical and research activities by helping health care providers to tailor preventive care according to the estimated mortality risk. Eventually, our study can help preventing under- and overuse of preventive care in the growing older population.”

12. Minor issues:

The authors might wish to make their study rationale stronger and more appealing. For instance, they might add a sentence or two to extrapolate the reasons why several of 16 prognostic indices were “fairly accurate to predict mortality, the authors concluded that none could be recommended for a widespread use”.

Authors’ response: We agree with the reviewer and have strengthened our Introduction section: “One major limitation is that none of these prognostic indices has been tested prospectively in various samples. Key is that their transportability in other populations is unknown and clinicians cannot use these indices with confidence across different groups of patients.” 

Reviewer 2 

Important topic, clear rationale of the study. I would like to congratulate the authors with a clearly written, and concise paper. I do have a few comments:

Authors’ response: We thank the reviewer for their comments below allowing us to improve our manuscript. 

13. -the statistical analyses are sophisticated, and can be considered state-of-the-art in the field of health outcome predictions

Authors’ response: Thank you.

14. -I do miss an independent population to assess external validity of the model

Authors’ response: We agree with the reviewer and have stated in the Discussion section: “One major limitation is the lack of external validation but we will explore opportunities to test the score in a different dataset of older multimorbid adults. We did however our best to assess internal validation by using the bootstrap method for internal validation that makes use of the entire sample.”

15. -The paper could be further strengthend by comparing performance of the model with other (simple) models that predict mortality. Is this model really more appropriate than those currently available?

Authors’ response: We are not sure to understand this comment, because we have compared our model with a simplified one and because there are no other models currently available. These points have been discussed in the Introduction and Discussion section of the paper.

16. -as the authors do acknowledge in the discussion, assessing performance in an extended time horizon beyond 1-year would be of interest

Authors’ response: We agree with the reviewer. As discussed in the Discussion section, the follow-up of our cohort study is still ongoing and we are currently working on extending the prognostic model to 3-year mortality risk: “Finally, prognostic information longer than 1-year mortality risk is needed. We will expand this model to 3-year mortality risk once the 3-year follow data collection is completed. [Gastens 2021]”

Comments from authors

During the revision process, we noticed a slight error in the calculation of the Charlson-Comorbidity-Index. We adapted the results accordingly. This did not change the findings in a substantial way or interpretation of the results. 

References

Moons, Karel GM, et al. "Transparent Reporting of a multivariable prediction model for Individual Prognosis or Diagnosis (TRIPOD): explanation and elaboration." Annals of internal medicine 162.1 (2015): W1-W73.

Royston, Patrick, and Willi Sauerbrei. Multivariable model-building: a pragmatic approach to regression anaylsis based on fractional polynomials for modelling continuous variables. Vol. 777. John Wiley & Sons, 2008.

Gastens, Viktoria, et al. "Development and validation of a life expectancy estimator for multimorbid older adults: a cohort study protocol." BMJ open 11.8 (2021): e048168.

---

## [Decision Letter · Decision Letter 1]

26 May 2022

PONE-D-22-03428R1Development and validation of a new prognostic index for mortality risk in multimorbid adultsPLOS ONE

Dear Dr. Gastens,

Thank you for submitting your manuscript to PLOS ONE. After careful consideration, we feel that it has merit but does not fully meet PLOS ONE’s publication criteria as it currently stands. Therefore, we invite you to submit a revised version of the manuscript that addresses the points raised during the review process.

Model performance must be thoroughly discussed.

We look forward to receiving your revised manuscript.

Kind regards,

Robert Daniel Blank, MD, PhD

Academic Editor

PLOS ONE

Journal Requirements:

Reviewers' comments:

Reviewer's Responses to Questions

**Comments to the Author**

1. If the authors have adequately addressed your comments raised in a previous round of review and you feel that this manuscript is now acceptable for publication, you may indicate that here to bypass the “Comments to the Author” section, enter your conflict of interest statement in the “Confidential to Editor” section, and submit your "Accept" recommendation.

Reviewer #1: (No Response)

2. Is the manuscript technically sound, and do the data support the conclusions?

Reviewer #1: Partly

3. Has the statistical analysis been performed appropriately and rigorously? 

Reviewer #1: Yes

4. Have the authors made all data underlying the findings in their manuscript fully available?

Reviewer #1: Yes

5. Is the manuscript presented in an intelligible fashion and written in standard English?

Reviewer #1: Yes

6. Review Comments to the Author

Reviewer #1: I would like to thank the authors for their efforts to address my concerns. There are however two issues that I appreciate their clarification.

1. Please explain and cite relevant references for your response “However, nomograms represent models with continuous variables. Due to the binary and categorical characteristics of our predictor variables, a nomogram analysis is not applicable.”. The authors might find in the current literature that the nomograms with binary or categorical predictor variables are common in many different research fields, including but not limited to oncology (https://www.ncbi.nlm.nih.gov/pmc/articles/PMC4465353/), bone (https://pubmed.ncbi.nlm.nih.gov/17370100/ ) and other research fields (https://www.ncbi.nlm.nih.gov/pmc/articles/PMC7906717/;
https://pubmed.ncbi.nlm.nih.gov/31132089/ ).

If the authors decide not to present the nomogram which they agree to be graphically intuitive, they might wish to add a sentence or two in the Discussion section to mention the nomogram analysis might be considered in the extended project.

2. The optimism corrected validation analysis using bootstrapping technique indicated both calibration slop and intercept significantly deviated from the null value of one (0.93; 95% CI: 0.92, 0.94) and zero (0.61; 0.56, 0.66) (Table 3A), respectively. The far less accurate calibration was noticed for the simplified model in Table 3B.

These findings indicated the significant suboptimal prediction accuracy of these prediction models as the risk estimates were considered too moderate (for the calibration slop) or the model significantly underestimated the predicted risk (for the intercept) (https://pubmed.ncbi.nlm.nih.gov/31842878/ ). Regardless of their importance and significant, these results have not been discussed thoroughly in the current manuscript.

The authors are expected to discuss sufficiently about the significantly suboptimal calibration of the prediction models. They might also wish to discuss the potential impact of this suboptimal prediction accuracy on the implementation of the model and future projects.

Additionally, the authors might wish to cross check the assumption for calculating the 95% CI of the calibration intercept given its upper limit of 1.24 (Table 3B).

3. It is statistically very hard to interpret the simplified prediction model had “slightly weaker discrimination power” as its C index (0.59; 95% CI: 0.58, 0.59) appeared to be significantly lower than the “full” prediction model (0.70; 0.69, 0.70). Similar results were found for both calibration intercept and slop. These metrics instead the simplified prediction model had significantly weaker prediction performance, in terms of both discrimination and calibration power than the full prediction model. As a result, the authors might also wish to revise the interpretation related to the simplified prediction model accordingly and discuss its potential implication further.

7. PLOS authors have the option to publish the peer review history of their article (what does this mean?). If published, this will include your full peer review and any attached files.

Reviewer #1: No

---

## [Author Response · Author response to Decision Letter 1]

8 Jul 2022

Reviewer 1

I would like to thank the authors for their efforts to address my concerns. There are however two issues that I appreciate their clarification.

1. Please explain and cite relevant references for your response “However, nomograms represent models with continuous variables. Due to the binary and categorical characteristics of our predictor variables, a nomogram analysis is not applicable.”. The authors might find in the current literature that the nomograms with binary or categorical predictor variables are common in many different research fields, including but not limited to oncology (https://www.ncbi.nlm.nih.gov/pmc/articles/PMC4465353/), bone (https://pubmed.ncbi.nlm.nih.gov/17370100/) and other research fields (https://www.ncbi.nlm.nih.gov/pmc/articles/PMC7906717/;
https://pubmed.ncbi.nlm.nih.gov/31132089/ ).

If the authors decide not to present the nomogram which they agree to be graphically intuitive, they might wish to add a sentence or two in the Discussion section to mention the nomogram analysis might be considered in the extended project.

Authors’ response: We agree with the reviewer and have added to the Discussion section “We will expand this model to 3-year mortality risk once the 3-year follow-up data collection is completed. [Gastens 2021] For this next research project, we might consider nomogram analysis as a graphically intuitive representation. [Balachandran 2015]”.

2. The optimism corrected validation analysis using bootstrapping technique indicated both calibration slop and intercept significantly deviated from the null value of one (0.93; 95% CI: 0.92, 0.94) and zero (0.61; 0.56, 0.66) (Table 3A), respectively. The far less accurate calibration was noticed for the simplified model in Table 3B.

These findings indicated the significant suboptimal prediction accuracy of these prediction models as the risk estimates were considered too moderate (for the calibration slop) or the model significantly underestimated the predicted risk (for the intercept) (https://pubmed.ncbi.nlm.nih.gov/31842878/). Regardless of their importance and significant, these results have not been discussed thoroughly in the current manuscript.

The authors are expected to discuss sufficiently about the significantly suboptimal calibration of the prediction models. They might also wish to discuss the potential impact of this suboptimal prediction accuracy on the implementation of the model and future projects. Additionally, the authors might wish to cross check the assumption for calculating the 95% CI of the calibration intercept given its upper limit of 1.24 (Table 3B).

Authors’ response: We agree with the reviewer and have added to our Discussion section: “This optimism-corrected validation analysis indicates that both calibration slope and intercept deviate from the null value of one (0.93; 95% CI: 0.92, 0.94) and zero (0.61; 95% CI: 0.56, 0.66) (Table 3A), respectively. The calibration slope evaluates the spread of the estimated risks and has a target value of 1. A slope < 1 suggests that estimated risks are too extreme, a slope > 1 suggests that risk estimates are too moderate. [Van Calster 2019] The calibration intercept, which is an assessment of calibration-in-the-large, has a target value of 0; negative values suggest overestimation, whereas positive values suggest underestimation. [Van Calster 2019] Therefore, our results indicate suboptimal prediction accuracy as the risk estimates were considered too extreme (for the calibration slope) and the model underestimating the predicted risk (for the calibration intercept).”.

Additionally, we checked the calculations for the 95% CI of the calibration intercept and confirm its upper limit of 1.24 (Table 3B).

3. It is statistically very hard to interpret the simplified prediction model had “slightly weaker discrimination power” as its C index (0.59; 95% CI: 0.58, 0.59) appeared to be significantly lower than the “full” prediction model (0.70; 0.69, 0.70). Similar results were found for both calibration intercept and slop. These metrics instead the simplified prediction model had significantly weaker prediction performance, in terms of both discrimination and calibration power than the full prediction model. As a result, the authors might also wish to revise the interpretation related to the simplified prediction model accordingly and discuss its potential implication further.

Authors’ response: We agree with the reviewer and have adapted our Discussion section to “Another limitation is that indices such as the Barthel-Index and the Charlson-Comorbidity-Index used as predictor may reduce the ease of use of the risk score at the point-of-care. We have therefore developed a simpler score without such variables. This simpler score showed reduced discriminative power compared to the full score. This could highlight the importance of taking the severity of comorbidities and functional impairment into account to improve risk prediction in older multimorbid people.”.

---

## [Decision Letter · Decision Letter 2]

11 Jul 2022

Development and validation of a new prognostic index for mortality risk in multimorbid adults

PONE-D-22-03428R2

Dear Dr. Gastens,

We’re pleased to inform you that your manuscript has been judged scientifically suitable for publication and will be formally accepted for publication once it meets all outstanding technical requirements.

Kind regards,

Robert Daniel Blank, MD, PhD

Academic Editor

PLOS ONE

Additional Editor Comments (optional):

Reviewers' comments:

Reviewer's Responses to Questions

**Comments to the Author**

1. If the authors have adequately addressed your comments raised in a previous round of review and you feel that this manuscript is now acceptable for publication, you may indicate that here to bypass the “Comments to the Author” section, enter your conflict of interest statement in the “Confidential to Editor” section, and submit your "Accept" recommendation.

Reviewer #1: All comments have been addressed

2. Is the manuscript technically sound, and do the data support the conclusions?

Reviewer #1: Yes

3. Has the statistical analysis been performed appropriately and rigorously? 

Reviewer #1: Yes

4. Have the authors made all data underlying the findings in their manuscript fully available?

Reviewer #1: Yes

5. Is the manuscript presented in an intelligible fashion and written in standard English?

Reviewer #1: Yes

6. Review Comments to the Author

Reviewer #1: I would like to thank the authors for their efforts to address my concerns and make the manuscript better.

7. PLOS authors have the option to publish the peer review history of their article (what does this mean?). If published, this will include your full peer review and any attached files.

Reviewer #1: No

---

## [Editor Report · Acceptance letter]

27 Jul 2022

PONE-D-22-03428R2 

Development and validation of a new prognostic index for mortality risk in multimorbid adults 

Dear Dr. Gastens:

I'm pleased to inform you that your manuscript has been deemed suitable for publication in PLOS ONE. Congratulations! Your manuscript is now with our production department. 

Kind regards, 

on behalf of

Professor Robert Daniel Blank 

Academic Editor

PLOS ONE